# Enzyme Inhibitors from Gorgonians and Soft Corals

**DOI:** 10.3390/md21020104

**Published:** 2023-01-31

**Authors:** Andrea Córdova-Isaza, Sofía Jiménez-Mármol, Yasel Guerra, Emir Salas-Sarduy

**Affiliations:** 1Ingeniería en Biotecnología, Facultad de Ingeniería y Ciencias Aplicadas, Universidad de Las Américas, Quito 170125, Ecuador; 2Grupo de Bio-Quimioinformática, Universidad de Las Américas, Quito 170125, Ecuador; 3Instituto de Investigaciones Biotecnológicas (IIB) “Dr. Rodolfo A. Ugalde”, Consejo Nacional de Investigaciones Científicas y Técnicas (CONICET), Buenos Aires B1650HMP, Argentina; 4Escuela de Bio y Nanotecnología (EByN), Universidad de San Martín (UNSAM), Buenos Aires B1650HMP, Argentina

**Keywords:** enzyme inhibitors, natural products, gorgonian, soft coral, enzyme kinetics, inhibitor characterization, STRENDA guidelines

## Abstract

For decades, gorgonians and soft corals have been considered promising sources of bioactive compounds, attracting the interest of scientists from different fields. As the most abundant bioactive compounds within these organisms, terpenoids, steroids, and alkaloids have received the highest coverage in the scientific literature. However, enzyme inhibitors, a functional class of bioactive compounds with high potential for industry and biomedicine, have received much less notoriety. Thus, we revised scientific literature (1974–2022) on the field of marine natural products searching for enzyme inhibitors isolated from these taxonomic groups. In this review, we present representative enzyme inhibitors from an enzymological perspective, highlighting, when available, data on specific targets, structures, potencies, mechanisms of inhibition, and physiological roles for these molecules. As most of the characterization studies for the new inhibitors remain incomplete, we also included a methodological section presenting a general strategy to face this goal by accomplishing STRENDA (Standards for Reporting Enzymology Data) project guidelines.

## 1. Introduction

Enzyme inhibitors are ubiquitous in nature and, as key regulators of enzyme activity, play pivotal roles in the physiology of all living forms. At a molecular level, endogenous inhibitors regulate enzymatic processes such as metabolism and nutrition, transport, signal transduction, DNA damage repair, apoptosis, cell differentiation, and cell cycle progression, among others. Therefore, many enzyme inhibitors are directly involved in critical events in the context of health and disease [1,2,3,4,5,6,7,8]. Natural enzyme inhibitors also target exogenous enzymes, such as those from invading or competing organisms. Pathogenic viruses, bacteria, fungi, and parasites usually depend on enzymes as virulence factors to guarantee successful colonization and survival. Additionally, digestive enzymes can be used as part of the offensive armamentarium of living organisms in highly crowded environments to avoid competition. Thus, endogenous enzyme inhibitors result in effective defensive immune or anti-predatory mechanisms against such molecular weapons [9,10]. Consequently, natural enzyme inhibitors have received notorious attention as a research subject and have found numerous biomedical [11,12], industrial [13], and environmental applications [14,15].

Marine environments house many different organisms which produce a huge diversity of bioactive molecules. In particular, they have proven to be a prolific source of novel enzyme inhibitors with unique functional features. Inhibitors for all major enzyme classes have been isolated and characterized from marine sources, including kinases [16,17,18,19], proteases [20], phospholipases [21], glucosidases [22], cholinesterases [23,24], topoisomerases [25], and DNA polymerases [26], among others. Accordingly, many of them have found interesting investigational, biomedical, and biotechnological applications [27,28,29]. The anthozoan sub-class Octocorallia, including over 3000 extant species of soft corals, gorgonian, and sea pens, has been considered a promising reservoir of unique natural products with unusual bioactivities [30]. Octocorals are widely distributed in all marine environments around the world and comprise three taxonomic orders: Helioporacea, Pennatulacea, and Alcyonaceae [30]. Although not strict in a taxonomic sense, the term “soft coral” is commonly applied to organisms in the Pennatulacea and Alcyonaceae orders with their polyps embedded within a fleshy mass of coenenchymal tissue [31]. Similarly, the term “gorgonian” is used to designate multiple species of the Alcyonaceae order producing a skeletal axis (or axial-like layer) composed of calcite and the proteinaceous material gorgonin [31]. Both soft corals and gorgonians have been found to be sources of numerous natural products with high structural diversity and interesting bioactivities, with terpenoids, steroids, and alkaloids as the most popular [4]. Although less represented in the scientific literature, several fascinating enzyme inhibitors have been also reported from these sources [32].

Due to their great diversity and potential, there is a sustained interest in identifying new enzyme inhibitors from these marine organisms. In the past decade, we worked actively on the prospection and characterization of protease inhibitory activities in the aqueous extracts of invertebrates from the Caribbean Sea [33,34,35,36]. Our interest focused on tight-binding protease inhibitors from the gorgonian *Plexaura homomalla*, and by transference, in related taxonomic classes such as soft corals. Thus, in this work, we decided to review recent progress in the identification and enzymological characterization of enzyme inhibitors from these marine organisms. From this literature analysis, protein-tyrosine phosphatase 1B (PTP1B) and the ubiquitin-proteasome system emerged as the enzymes most frequently targeted by novel inhibitors, while terpenes and terpenoids predominated among inhibitors. We also observed that, in contrast with the detailed structural studies present in almost all the studies, the enzymological characterization was insufficient or absent for many of the newly reported inhibitors. Critical aspects of inhibition, such as reversibility, time dependency, inhibition type, and the value for the inhibition thermodynamic constant K_i_ [37], must be properly determined so that these bioactive molecules find applications according to their potential. Considering that most novel inhibitors are indeed identified in laboratories not specialized in enzymology, we also included in this review a final section with strategic recommendations on how to conduct such kind of characterization, using STRENDA (Standards for Reporting Enzymology Data) guidelines as a methodological framework [37,38,39]. 

## 2. Enzyme Inhibitors Isolated from Gorgonians and Soft Corals

Figure 1 graphically resumes relevant statistics on the collected information describing enzyme inhibitors from soft corals and gorgonians. We found and processed over 50 scientific reports, published between 1974 and 2022, which contained 127 inhibitor molecules (Appendix A). Concerning their origin, soft corals were the preferred source for the prospection of new inhibitors in this period (Figure 1A), accounting for about 64% of the reviewed papers. This tendency was confirmed in the last 5 years when the proportion increased to 81% (Figure 1B). Considering that 206 original papers from this field (i.e., search terms: “natural product” AND “soft coral” OR “gorgonian” NOT “Review”) were deposited in PUBMED in the period 1974–2022, the class of enzyme inhibitors seemed to be a predominant one, representing nearly 25% of the total scientific production.

For classification purposes, the target enzymes for the reported inhibitors were grouped into four major classes (i.e., hydrolases, oxidoreductases, transferases, and translocases) according to the reaction type they catalyze. Hydrolases and transferases were, by far, the most abundant among them (Figure 1C and Table 1). 

Lastly, we grouped inhibitors by targets and tabulated their structures, potencies (predominantly as half-maximal inhibitory concentrations or IC_50_ values), and source organism. Inhibitors were classified into six major classes (i.e., terpenes and terpenoids, steroids, peptides, eicosanoids, alkaloids, and hydroquinones), and showed relatively low structural diversity. As expected, more than half of the inhibitory molecules identified in this period from soft corals and gorgonians were terpenes and terpenoids (Figure 1D), two classes that have historically dominated the literature on coral-derived natural products. Surprisingly, the functional characterization of novel enzyme inhibitors was biased toward cellular toxicity studies and not toward the in-depth understanding of their inhibitory activity. In the few cases where further enzymological information was available (e.g., inhibition mechanism, the value for the thermodynamic constant K_i_, or target specificity), we included it within the text. 

### 2.1. Hydrolases

#### 2.1.1. Protein Tyrosine Phosphatase 1B (PTP1B) (EC 3.1.3.48)

The enzyme PTP1B is known as a target in the search for new drugs in the treatment of type 2 diabetes, obesity, and breast cancer [40,41,42]. This enzyme is one of the most used targets in the evaluation of marine compounds as enzyme inhibitors from soft corals. Twenty-eight molecules from seven soft coral and gorgonian species have been reported as PTP1B inhibitors (Table 2 and Figure 2). Most of these compounds are diterpenes or diterpenoids, with a few cases of steroids. The soft coral *Sarcophyton trocheliophorum Marenzeller* is the source of the highest number of isolated PTP1B inhibitors, such as the capnosane-type diterpenes Sarsolides A (**1**) and B (**2**) [43], the cembrane diterpenoids sarcophytonolide N (**3**), sarcrassin E (**4**), cembrene C (**5**), 4*Z*,12*Z*,14*E*-sarcophytolide (**6**), ketoemblide (**7**) [44], and the diterpenoid Methyl sarcotroate B (**8**) (Table 2 and Figure 2) [45]. The inhibitory results from capnosane-type diterpenes suggest that the presence of an exomethylene Δ10 (17) of the capnosane skeleton may play an important role in inhibitory activity [43]. In a similar way, some preliminary structure–function relationships could be proposed for the cembrane diterpenoids. The most potent molecules, sarcophytonolide N and sarcrassin E, have a dienoate moiety at C-1 through C-18, while the presence of an α-β-unsaturated lactone in sarcrassin E seems to be not essential for the inhibition. On the other hand, the presence of the methyl ester group at C-18 in sarcophytonolide N and sarcrassin E increases the inhibitory activity. The lack of cytotoxicity against human cell lines A-549 and HL-60 makes these compounds promising hits for the development of a new class of PTP1B inhibitors [43]. Methyl sarcotroates B is the only PTP1B inhibitor isolated from a marine organism containing a hydroperoxide group [43].

PTP1B inhibitors have also been isolated from several species of the soft coral genus *Sinularia*. The cembranoids sinulin D (**9**) and (1*R*,3*S*,4*S*,7*E*,11*E*)-3,4-epoxycembra-7,11,15-triene (**10**), together with the sesquiterpenoid 15-hydroxy-α-cadinol (**11**), were isolated from *Sinularia* sp. [46]. These three terpenoids showed IC_50_ values in the millimolar range, which can be considered a weak inhibitory activity [46]. The prenyleudesmane-type diterpene sinupol (**12**), and (1*R*,4*aR*,6*S*,8*aS*)-6-((2*E*,4*E*)-6-hydroxy-6-methylhepta-2,4-dien-2-yl)-4,8*a*-dimethyl-1,2,4*a*,5,6,7,8,8*a*-octahyronaphthalen-1-ol (**13**) have been isolated from the soft coral *Sinularia polydactyla* [47]. Furthermore, the capnosane-type diterpenoid sinulacetate (**14**) and the nardosinane-type sesquiterpenoid linardosinene I (**15**) were found in *Sinularia polydactyla* and *Litophyton nigrum* [48], respectively. Additionally, the guaiane-type sesquiterpenes Molestins C (**16**) and D (**17**), together with the furanosesquiterpene (5´*Z*)-5-(2′,6′-Dimethylocta-5′,7′-dienyl)-furan-3-carboxylic acid (**18**), were isolated from *Sinularia* cf. *molesta* [49]. On the other hand, the steroids (3β,4α,5α,8β)-4-methylergost-24(28)-ene-8-ol-3-monoacetate (**19**), (3β,4α,5α)-4-methylergost-24(28)-ene-3-ol, ergost-4,24(28)-diene-3-one (**20**), and ergost-4,24(28)-diene-3-one (**21**) were isolated from *Sinularia depressa* [50], while the polyhydroxylated steroid 7α-hydroxy-crassarosterol A (**22**) was found in *Sinularia flexibilis* [51]. A potential role of substitution of hydroxyl groups in positions 3 and 8 is proposed, based on the results obtained from steroids with the same scaffold isolated from *Sinularia depressa*.

Several new lipidyl pseudopteranes (**23**–**28**) with inhibitory activity towards PTP1B were found in the gorgonian *Pseudopterogorgia acerosa* [52]. These compounds were reported as the first pseudopterane diterpenes with a fatty acid moiety and, based on the enzymatic assays performed with several enzymes of the protein tyrosine phosphatase family, they have been described as selective inhibitors of PTP1B.

#### 2.1.2. Acetylcholinesterase (EC 3.1.1.7)

The enzyme acetylcholinesterase is a validated drug target for the treatment of Alzheimer´s disease [53], with several inhibitors approved by the FDA for its clinical use [54]. All the acetylcholinesterase inhibitors described from soft coral and gorgonians are diterpenes or sesquiterpenes, with little or no quantitative data about their inhibition. The most potent and best-characterized inhibitors are the cembranes asperdiol (**29**) and 14-acetoxycrassine (**32**), isolated from the soft coral *Eunicea knighti* and the gorgonian *Pseudoplexaura porosa*, respectively (Table 3 and Figure 3) [55]. Other cembranes, with acetylcholinesterase inhibitory activity, were found in *Eunicea knighti* and *Pseudoplexaura flagellosa*; but no further kinetic characterization was performed [55]. Additionally, the diterpene sarcophine (**33**), produced by *Sarcophyton glaucum*, was described as a competitive inhibitor of acetylcholinesterase, although no IC_50_ or K_i_ value was reported [56]. In the same study, it was suggested that sarcophine (**33**) inhibits acetylcholinesterase through the formation of an adduct with the free cysteines of the enzyme. Other compounds, that have been classified as weak acetylcholinesterase inhibitors are the sesquiterpenoid sinuketal (**34**) and the cembranoid crassumolide E (**35**), which were isolated from *Sinularia* sp. and *Lobophytum* sp., respectively [46,57]. It should be noted that crassumolide E (**35**) was considered a weak inhibitor in comparison with galanthamine, although the minimal concentration of crassumolide E needed to inhibit the enzyme was in the nanomolar range.

#### 2.1.3. HIV-1 Protease (EC 3.4.23.16)

HIV-1 protease is a well-established drug target for the treatment of HIV-AIDS [62,63]. In fact, several inhibitors of this enzyme are used in highly active antiretroviral therapy used for the treatment of this disease [64]. Several molecules with inhibitory activity against the protease of HIV-1 have been identified from the soft coral *Litophyton arboreum* (Table 3 and Figure 3). The structural diversity among the compounds alismol (**36**), 7β-acetoxy-24-methylcholesta-5-24(28)-diene-3,19-diol (**37**), erythro-N-dodecanoyl-docosasphinga-(4*E*,8*E*)-dienine (**38**), sarcophytol M (**39**), and chimyl alcohol (**40**) is significant, with the presence of sesquiterpenes, diterpenes, glyceryl ethers, and steroids [58]. The most potent compounds are 7β-acetoxy-24-methylcholesta-5-24(28)-diene-3,19-diol (**37**) and erythro-N-dodecanoyl-docosasphinga-(4*E*,8*E*)-dienine (**38**) with remarkably similar potency against the HIV-1 protease. 

#### 2.1.4. Elastase (EC 3.4.21.11)

Elastases are serine proteases belonging to the chymotrypsin-like family. Human neutrophil elastase (HNE) is one of the best-characterized elastases, as it plays important roles in inflammation, the elimination and degradation of extracellular pathogens, and the activation of other proteases [65]. As a key inflammation mediator, the discovery and development of new HNE inhibitors have received significant interest [66]. 

Kunitz-like peptides 1–3 (PcKuz 1–3), isolated from *Palythoa caribaeorum*, are a few of the peptide-nature molecules described as enzyme inhibitors from soft corals [59]. However, the weak inhibitory activity against the serine proteases trypsin and elastase, and the results of toxicity tests with zebrafish larvae, lead the authors of this study to suggest that these peptides should be considered Kunitz-type neurotoxins rather than protease inhibitors. 

Three prostaglandins (**41**–**43**) with inhibitory activity against elastase were isolated from the gorgonian *Plexaura homomalla* (Table 3 and Figure 3) [60]. These compounds showed similar potency, in terms of inhibition, in the kinetic assays performed in this study, but no IC_50_ or K_i_ values were reported. Based on the results of the kinetics assays, it seems that the presence of the carboxyl group is essential for elastase inhibition, since the molecules with an acetyl ester in this position showed no inhibition on the enzyme [60]. 

#### 2.1.5. 3CLpro Enzyme (EC 3.4.22.69)

The main protease of severe acute respiratory syndrome coronavirus is critical for the replication cycle of these viruses [67]. The main protease of SARS-CoV-2 is a 3CLpro cysteine protease that performs the cleavage of 12 nonstructural proteins of the virus. Inhibitors of this protease have proved to be effective in the inhibition of viral replication in cell-based assays [68]. An inhibitor of the SARS-CoV-2 3CLpro protease, described as a cyclized merosesquiterpenoid with a new carbon scaffold and composed by a highly substituted chromene core, was found in *Duva florida* (Table 3 and Figure 3) [61]. This compound, named Tuaimenal A (**44**), showed no inhibitory activity against other cysteine proteases (*Fasciola hepatica* cathepsin L1 and L3, and human cathepsin L) or serine proteases (trypsin, chymotrypsin, and thrombin), suggesting a particular specificity towards 3CLpro protease. Cell-based studies have shown that Tuaimenal A (**44**) has low toxicity in cells that are sensitive to other protease inhibitors, highlighting the potential of this molecule as a hit in the development of new anti-coronavirus drugs. 

#### 2.1.6. Ubiquitin-Proteasome System (EC 3.4.25.1)

The ubiquitin–proteasome system plays a critical role in the maintenance of proteome homeostasis. Therefore, proteasome is considered an important target for the development of drugs for the treatment of diseases such as neurodegenerative disease, immune-related disease, and cancer [69]. Several inhibitors of the ubiquitin–proteasome system have been identified using a cell-based, high-content assay based on the measurement of aggregations of ubiquitinated proteins. Among these inhibitors are the cembrane-based molecules Sarcophytonin A (**45**) and Laevigatol A (**46**), which were isolated from the soft coral *Sarcophyton trocheliophorum;* as well as Sarcophytoxide (**47**) and Sarcophine (**33**), isolated from *Sarcophyton ehrenbergi* (Table 4 and Figure 4) [70]. Sarcophytonin A (**45**) and Sarcophine (**33**) seem to target the 19S proteasome, based on the results obtained by comparison to the mechanistic action of known proteasome inhibitors. All four compounds showed negligible cytotoxicity (ED_50_ > 25 µg/mL) to human HEK293T cells. In addition, the dolabellane-based compounds clavinflol C (**48**), stolonidiol (**49**), stolonidiol-17-acetate (**50**), and clavinflol B (**51**), and the secosteroid-based molecules 3β,11-dihydroxy-24-methyl-9,11-secocholest-5-en-9,23-dione (**52**), and 3β,11-dihydroxy-24-methylene-9,11-secocholest-5-en-9,23-dione (**53**) were purified from *Clavularia flava* [71]. These six molecules have shown less cytotoxicity than the known proteasome inhibitors bortezimid and MG132.

A diterpenoid (**54**) found in the gorgonian *Pseudopterogorgia acerosa* inhibits the chymotrypsin-like activity of the ubiquitin–proteasome system, with no inhibition of its caspase-like activity [72]. 

Punaglandins are highly functional cyclopentadienone and cyclopentenone prostaglandins, chlorinated at the endocyclic R-carbon position. A group of these molecules, isolated from the soft coral *Telesto riisei,* were used to investigate the inhibition mechanism previously proposed for dienone prostaglandins [76]. In this study, the punaglandins PNG 2 (**55**), PNG 3 (**56**), PNG 4 (**57**), Z-PNG 4 (**58**), and PNG 6 (**59**) showed inhibitory activity against the ubiquitin isopeptidase using in vitro and in vivo assays (Table 4 and Figure 4) [73]. The use of these punaglandins demonstrates that the presence of chloride in endocyclic carbon increases the inhibitory activity in comparison to unchlorinated prostaglandins. Additionally, the results obtained suggest that the inhibitory activity of the prostaglandins is related to the olefin-ketone conjugation and the reactivity of the endocyclic carbon [73].

#### 2.1.7. Phosphodiesterase-4 (EC 3.1.4.53)

The phosphodiesterases belong to a family of enzymes that catalyze the hydrolysis of the second messengers cyclic adenosine monophosphate (cAMP) and guanosine monophosphate (cGMP) [77]. Particularly, phosphodiesterase-4 is a promising drug target for diverse diseases including inflammatory, asthma, and chronic obstructive pulmonary disease [78]. Several prostaglandins showing inhibitory activity against phosphodiesterase-4 were isolated from the soft coral *Sarcophyton ehrenbergi*. A number of them were novel, such as compounds sarcoehrendin B (**60**), D (**61**), F (**62**), H (**63**), and J (**64**), while others, including 9α,15α-diacetoxy-11α-hydroxy-5*Z*,13*E*-prostadienoic acid methyl ester (**65**), (5*Z*,9α,11α,13*E*,15*S*)-11,15-bis(acetoxy)-9-hydroxyprosta-5,13-dien-1-oic acid methyl ester (**66**), and 9,11,15-triacetoxy PGF2α methyl ester (**67**), were already known molecules (Table 4 and Figure 5) [74]. Of these compounds, 5 showed IC_50_ values lower than 10 µM, while the inhibitory potency for 9,11,15-triacetoxy PGF2α methyl ester (**67**) was comparable to that of the positive control (rolipram). Based on the results obtained in this study, several structure–function relationships were proposed. For instance, the esterification at the OH-15 seems to be essential for a strong inhibitory activity, while the acetylation of OH-9 or OH-11 produces only a minor increase in activity [74].

Moreover, six tetraprenylated alkaloids found in the gorgonian *Echinogorgia pseudossapo* showed inhibitory activity towards phosphodiesterase-4 [75]. These compounds are the melonganenones L–Q (**68**–**73**), with melonganenones L (**68**) and Q (**73**) exhibiting IC_50_ values of 8.5 and 20.3 µM, respectively (Table 4 and Figure 5). All six tetraprenylated alkaloids also showed inhibition against the phosphodiesterases PDE5A and PDE9A, but with weaker potency than towards PDE4 [75].

#### 2.1.8. α-Glucosidase (EC 3.2.1.207)

α-Glucosidase is a key enzyme in carbohydrate metabolism, which regulates blood glucose. Through controlling postprandial glucose levels, this enzyme is a validated target for the treatment of type 2 diabetes, with several available inhibitors in the market [79,80]. Two cembranoid compounds with inhibitory activity towards α-glucosidase, sinulacrassin B (**74**) and S-(+)-cembrane A (**75**), were isolated from the soft coral *Sinularia crassa* (Table 5 and Figure 6) [81]. Both compounds showed higher inhibitory activity than 1-deoxynojirimycin, the positive control used in the enzymatic assay. In addition, these cembranoids were nontoxic towards human normal hepatocytes (LO2), with IC_50_ higher than 100 µM, providing a new scaffold for the development of anti-diabetes drugs.

#### 2.1.9. Histone Deacetylase 6 (HDAC6) (EC3.5.1.98)

Histone deacetylases are enzymes that remove acetyl groups from histone and non-histone proteins, altering their stability and activity. Histone deacetylase 6 is a member of the Class IIb subfamily that participates in several biological processes including cell motility [86], cell survival [87], protein degradation [88], and immunoregulation [89]. A new diterpene, named methyl (1*S*, 4*S*, 5*R*, 9*S*9-4-((*Z*)-4-(3,3-dimethyloxiran-2-yl-)-1-hydrioxybut-2-en-2-yl)-1-methyl-6-methylene-10-oxabicyclo[7.1.0]decane-5-carboxylate (**76**), was found in the soft coral *Xenia elongata* (Table 5 and Figure 6) [82]. This molecule selectively inhibits histone deacetylase 6, with no detectable inhibitory activity against deacetylases from class I (HDAC1, HDAC2, HDAC3, and HDAC8) and IIA (HDAC4, HDAC5, HDAC7, and HDAC9). 

#### 2.1.10. Phospholipase A2 (PLA2) (EC 3.1.1.4)

Phospholipases A2 are esterases that cleave phospholipids and release fatty acids and lysophospholipids. These enzymes are considered as promising targets in the treatment of cancer, inflammation, and atherosclerosis, among other diseases [90,91,92]. Several PLA2 inhibitors have been found in the gorgonian *Euplexaura anastomosans*. These molecules, described as farnesylhydroquinone glycosides, were named Euplexides [83,84]. The euplexides A (**77**), B (**78**), F (**79**), and G (**80**) exhibited inhibitory activities against PLA2 ranging from 47 to 71% at 50 µg/mL, but no further kinetic characterization has been performed. Additionally, the steroids 7*α*,8*α*-epoxy-3*β*,5*α*,6*α*-trihydroxycholestane (**81**) and 24-methyl-7*α*,8*α*-epoxy-3*β*,5*α*,6*α*-trihydroxycholest-22-ene (**82**) have been isolated from the gorgonian *Acabaria undulata*, both with inhibitory activity towards PLA2 (Table 5 and Figure 6) [85]. 

### 2.2. Oxidoreductases

#### 2.2.1. 5α-Reductase (EC 1.3.1.22)

The enzyme 5α-Reductase converts testosterone into the more potent androgen dihydrotestosterone. Inhibition of 5α-Reductase is a promising strategy in the treatment of benign hyperplasia and male pattern baldness [93,94]. Several soft steroidal molecules with inhibitory activity towards 5α-Reductase have been isolated from soft corals of the Adaman and Nicobar Islands. Some of these compounds were isolated from more than one species, such as the steroids 24-methylenecholest-5-ene-3β,7β, 16β-triol-3-O-α-l-fucopyranoside (**83**), that was found in three soft corals: *Sinularia crassa Tixier–Durivaul, Sinularia gravis Tixier–Durivault, Sinularia* sp., and 24-methylenecholest-5-ene-3β,7β, 16β-diol-3-O-α-l-fucopyranoside (**84**), isolated from *Sinularia* sp. and *Cladiella* sp. [95]. On the other hand, the molecules (24S)-24-methylcholestane-3β,5α,6β,7β-tetrol (**85**), and (24S)-24-methylcholestane-3β,5α,6β,25-tetrol (**86**) were isolated from *Lobophytum crassum* and *Lobophytum* sp., respectively [95]. In a different study, the diterpenoid lemnabourside (**87**) was identified in the soft coral *Nephthea chabroli* [96], displaying weak inhibitory activity against 5α-Reductase (Table 6 and Figure 7) [96].

#### 2.2.2. Cytochrome P450 1A (EC 1.14.14.1)

Cytochrome P450 enzymes are a superfamily with high versatility in drug metabolism, detoxification of xenobiotics, and biosynthesis of endogenous compounds [99]. Particularly, cytochrome P4501A is one of the most important enzymes involved in the tumorigenesis induced by environmental pollution [100]. Therefore, this enzyme has been considered as an attractive target for the development of anti-cancer drugs [99]. The cembranoids 12(S)-hydroperoxylsarcoph-10-ene (**88**), 8-epi-sarcophinone (**89**), and ent-sarcophine (**90**) were isolated from the Red Sea soft coral Sarcophyton glaucum (Table 6 and Figure 7) [97]. The results of the inhibition assays with these compounds revealed some interesting structure–function relationships. For instance, the differences obtained for the inhibitory potency for ent-sarcophine (**90**) and sarcophine (**33**) suggest that the configurations of atoms C1 and C6 are essential for the inhibition of the cytochrome P450 1A [97]. 

#### 2.2.3. Tyrosinase (EC 1.14.18.1)

Tyrosinase is a key enzyme which catalyzes a rate-limiting step in melanin synthesis, and the downregulation of its activity constitutes the most prominent approach for the development of melanogenesis inhibitors [101]. Thus, tyrosinase inhibitors are considered as promising candidates for the development of skin whitening agents [101]. The tyrosinase inhibitor 4-(phenylsulfanyl)butan-2-one (**91**) is isolated from the Formosan soft coral *Cladiella australis* (Table 6 and Figure 7). The kinetic characterization of this compound showed that it acts as a noncompetitive inhibitor of the mushroom tyrosinase, with a K_i_ value of 3.45 × 10^−5^ M. Additionally, in vitro cell-based assays revealed that 4-(phenylsulfanyl)butan-2-one (**91**) has low cytotoxicity towards several human cell lines [98].

### 2.3. Transferases

#### 2.3.1. Tyrosine Kinase p56^lck^ (TK) (EC 2.7.10.2)

The tyrosine kinase p56^lck^ is a lymphocyte-specific protein tyrosine kinase that is a member of the Src family of non-receptor protein kinases [102]. This kinase is involved in the phosphorylation of several intracellular signaling proteins such as protein kinase C, phosphoinositide 3-kinase, and Zeta-chain-associated protein kinase 70 [103]. 

Since Tyrosine kinase p56^lck^ participates in T cell proliferation and differentiation, its inhibitors could be used in the treatment of several inflammatory and autoimmune disorders [103]. Two sterols, with inhibitory activity towards tyrosine kinase p56lck were isolated from the soft coral *Capnella lacertiliensis*: 12β-acetoxyergost-5-ene-3β,11β,16-triol (**92**), and 11β-acetoxyergost-5-ene-3β,12β,16-triol (93) (Table 7 and Figure 8) [104].

#### 2.3.2. IKKbeta Kinase (EC 2.7.11.10)

IKKbeta is one of the two catalytic units that compose the kinase complex IkB [108]. IKKbeta is involved in nuclear factor-KB signaling, and hence in the pathogenesis and progression of inflammatory diseases [109]. The cembranoids 3,4-epoxy,13-oxo,7*E*,11*Z*,15-cembratriene (**94**), and 3,4-epoxy,13-oxo,7*E*,11*E*,15-cembratriene (**95**) were isolated from the soft coral *Sarcophyton* sp. and both showed inhibitory activity against the IKKbeta kinase (Table 7 and Figure 8) [105]. In the same study, the carotenoid astaxanthin (**96**) was isolated from the gorgonian *Subergorgia* sp., and it showed inhibitory activity towards the IKKbeta kinase [105]. However, it is likely that astaxanthin is not produced by *Subergorgia* sp., but instead acquired from a marine bacteria or algae [110].

#### 2.3.3. Epidermal Growth Factor Receptor Kinase (EGFR) (EC 2.7.10.1)

EGFR is a member of the epidermal growth factor receptor family. It was the first member of this family with supporting evidence linking its overexpression with cancer [111]. This relationship has been established for several types of cancer, including laryngeal [112], esophageal [113], and non-small cell lung cancer [114], among others. Thus, EGFR inhibitors are considered as attractive candidates for developing anticancer drugs. The pachycladin A (**97**) is a diterpenoid isolated from the soft coral *Cladiella pachyclados* and was able to inhibit the kinase enzymatic activity of the EGFR kinase 8 (Table 7 and Figure 8) [106]. The inhibition of pachycladin A (**97**) seems to be selective towards EGFR kinase since no inhibition was detected against the two other kinases from the EGFR family (HER2 and HER4). 

#### 2.3.4. Protein Kinase C (PKC) (EC 2.7.11.13)

Protein kinase C is a family of enzymes involved in cell signaling pathways that mediates critical events like cellular proliferation and gene expression regulation [115]. For these reasons, PKC is considered a key target for the treatment of various diseases, including cancer, and neurological and cardiovascular disorders [116,117,118]. Three new secosterols (**98**–**100**) showing PKC inhibitory activity in the micromolar range were isolated from the gorgonian *Pseudopterogorgia* sp. [107]. In this work, IC_50_ values ranged from 12 to 50 µM, but no individual values were reported for these compounds (Table 7 and Figure 8). Considering the role of PKC in inflammatory and proliferative processes, all these three molecules were evaluated in cell cultures and showed antiproliferative properties. For the compound (2*S*,4*aS*,5*S*,6*S*,8*aS*)-6-hydroxy-2-((1*S*,2*R*,3*R*)-2-(2-hydroxyethyl)-2-methyl-3-((2*R*,5*R*,*E*)-4,5,6-trimethylhept-3-en-2-yl)cyclopentyl)-5,8*a*-dimethyloctahydronaphthalen-1(2H)-one (**99**), anti-inflammatory properties were also confirmed. 

#### 2.3.5. Human Tumor-Related Protein Kinases

Several 9,10 secosteroids were isolated from the gorgonian *Astrogorgia* sp. and assessed against 16 different human tumor-related protein kinases [119]. The kinases tested were AKT1 (RAC-alpha serine/threonine-protein kinase), ALK (Anaplastic lymphoma kinase), ARK5 (AMPK-related protein kinase 5), Aurora-B, AXL (AXL receptor tyrosine kinase), FAK (Focal adhesion kinase), IGF-1R (Insulin-like growth factor 1 receptor tyrosine kinase), MEK1wt (MAP kinase 1), METwt (MET receptor Tyrosine Kinase), NEK2 (NIMA-related Kinase 2), NEK6 (NIMA-related Kinase 6), PIM1 (Serine/threonine-protein kinase PIM-1), PLK1 (Serine/threonine-protein kinase PLK1), PRK1 (Serine/threonine-protein kinase N1), SRC (Proto-oncogene tyrosine-protein kinase Src), and VEGF-R2 (VEGFR2 receptor tyrosine kinase). The compounds calicoferols A (**101**) and E (**102**), and 24-exomethylenecalicoferol E (**103**), 9β-hydroxy-9,10-secosteroid astrogorgol F (**104**), and 9α-hydroxy-9,10-secosteroid astrogorgiadiol (**105**) showed significant inhibition against kinases LK, AXL, FAK, IGF1-R, MET wt, SRC, and VEGF-R2 (Table 8 and Figure 9) [119]. On the other hand, 9,16 di-oxygenated molecules, such as calicoferols I and B, showed weak inhibition (IC_50_ > 100 µM) against these kinases, pointing to the oxygenation at the C-16 as the cause for decreased inhibitory activity. The authors of this work concluded that the 9-oxygenated 9,10-secosterol nucleus is the basis of the inhibitory activity of these compounds towards the tested kinases. Additionally, the tetraprenylated alkaloid malonganenone D (**106**), isolated from the gorgonian *Euplexaura robusta*, has shown inhibitory activity against the MET Receptor Tyrosine Kinase, or c-Met [120]. This compound had been also previously isolated from the gorgonian *Euplexaura nuttingi* [121].

#### 2.3.6. Casitas B-Lineage Lymphoma Proto-Oncogene B (Cbl-b) (E3-Ubiquitin Ligase) (EC 2.3.2.27)

Casitas B-lineage lymphoma proto-oncogene B (Cbl-b) is an E3 ubiquitin ligase that acts as an important regulator of the immune response [125]. Targeting this enzyme is a promising approach for the treatment of autoimmune diseases and cancer [126,127]. The Sinularamides A−G (**107**−**113**), isolated from the soft coral *Sinularia* sp., are a group of diterpenoids that inhibits Cbl-b enzyme [122]. The most potent of these compounds was the sinularamide C (**109**), displaying an IC_50_ value in the low micromolar range (Table 8 and Figure 9). 

#### 2.3.7. Farnesyl Protein Transferase (EC 2.5.1.58)

The farnesyl protein transferase is an enzyme that adds a 15-carbon isoprenoid to a cysteine amino acid of Ras protein [128]. Oncogenic mutations of Ras protein are quite common in cancer, making the protein that is involved in its post-transcriptional modification a promising way to block Ras signal transduction [129,130]. The molecule 1*aR*,4*E*,8*E*,11*S*,11*aR*,14*aS*,14*bS*)-1*a*,5,9,14*a*-tetramethyl-12-methylene-13-oxo-1*a*,2,3,6,7,10, 11,11*a*,12,13,14*a*,14b-dodecahydrooxireno[2′,3′:13,14]cyclotetradeca[1,2-*b*]furan-11-yl acetate (**114**), with inhibitory activity towards recombinant human farnesyl protein transferase, was isolated from the soft coral *Lobophytum cristagalli* [123]. Kinetic assays suggest that this inhibitor competes with the Ras protein, substrate of the farnesyl protein transferase. The apparent K_i_ value determined in this kinetic characterization was 0.17 µM. However, the same inhibitor is noncompetitive with respect to the farnesyl pyrophosphate substrate. In the same study, this compound also inhibited the enzyme geranyl protein transferase-1, closely related to the farnesyl protein transferase, but with lower potency (IC_50_: 5.3 µM) [123]. 

#### 2.3.8. Glutathione S-Transferase (EC 2.5.1.18) 

Glutathione S-transferases (GST) are a family of enzymes that catalyze the conjugation of glutathione to a wide number of xenobiotics, making them more hydrophilic and facilitating their elimination [131]. Although GST protects the cell from toxic products, it also reduces the effectiveness of certain anticancer drugs [132]. The use of inhibitors of GST could, therefore, increase the sensitivity of tumor cells to anticancer drugs [133].

Crude and aqueous extracts of the gorgonian *Plexaura homomalla* have shown inhibitory activity towards the GST enzyme from the gastropod *Cyphoma gibbosum*. The structural analysis of these extracts revealed the presence of several series of prostaglandins (**115-119**). The use of commercially available prostaglandins that represents the diversity of classes found in *P. homomalla* confirmed the capacity of the molecules 15(*S*)-PGA2 (**115**), 15(*R*)-15-methyl PGA2 (**116**), 15(*S*)-PGE2 (**117**), 15(*R*)-PGE2 (**118**), and 15(*S*)-PGF2a (**119**) to inhibit GST (Table 8 and Figure 9). The results of the enzymatic assays revealed that compounds with the cyclopentenone ring showed the highest inhibitory activity [124]. 

### 2.4. Translocases

#### H (+)-Pyrophosphatase (EC 7.1.3.1)

Vacuolar H(+)-pyrophosphatase is an electronic proton pump present in most land plants and in other organisms such as algae, bacteria, protozoa, and archaebacteria [134]. In plants, this enzyme participates in the cytosolic hydrolysis of PPi and in vacuole acidification [135]. Specific inhibitors of the enzyme would be useful for investigating its physiological role and its biochemical characteristics. Acylspermidines A (**120**), B (**121**), C (**122**), D (**123**), and E (**124**) (Figure 10) from the soft coral *Sinularia* sp. inhibit plant vacuolar H(+)-pyrophosphatase from *Vigna radiata cv. Wilczek* [136]. These acylspermidine derivates showed a noncompetitive inhibition toward the vacuolar H(+)-pyrophosphatase. However, these compounds did not inhibit the vacuolar H+-ATPase, plasma membrane H+-ATPase, mitochondrial ATPase, or cytosolic PPase, suggesting a specific inhibitory activity against the vacuolar H(+)-pyrophosphatase. 

## 3. Walking the Whole Path to Describe an Enzyme Inhibitory Activity: What Next, after Screening and Structure Elucidation?

Previous sections reveal that, despite the number of enzyme inhibitors identified in the last five decades from gorgonians and soft corals, none of them, to date, possess a complete enzymological characterization. Remarkably, in many of these works, not even an IC_50_ value is calculated, but only an inhibition percentage at a fixed inhibitor concentration. This trend, which is extendable to other natural sources, reflects both (i) the lack of a uniform procedure for the enzymological characterization of inhibitors from marine organisms and (ii) the challenge that this goal represents for users unfamiliar with enzymology. 

The STRENDA project provides guides for the minimal information set required for reporting enzymological data [38,39,137]. For inhibitors, report requirements include reversibility and time-dependency, inhibition type, Ki value, and the method used for its estimation [37]. Using these parameters as a reference, many reports of enzyme inhibitors, and particularly those from marine sources, appear noticeably incomplete. Although several factors might be involved, we believe that the scarcity of fully characterized inhibitors might be due to the lack of expertise of many laboratories in studying this bioactivity. Therefore, we believe that a brief description of critical steps might encourage the study of a higher number of novel enzyme inhibitors from this prominent source. 

Figure 11 presents a general strategy for characterizing new enzyme inhibitors independent of their nature and origin, adapted from the one proposed by Copeland [138]. Although it admits some flexibility in the order of steps, the selected approach is, in our opinion, intuitive and the most convenient to implement in unexperienced laboratories. In brief, the strategy consists of five steps: (1) obtaining a preliminary assessment of inhibitory potency (i.e., the range of molar concentrations in which it effectively inhibits the target enzyme) by determining half-maximal inhibitory concentration (IC_50_); (2) determining inhibition reversibility and temporal dependency; (3) elucidation of inhibition type; (4) estimation of K_i_ value; and (5) a specificity validation step.

Determining a preliminary estimate of inhibitor potency (Step 1) is critical, as this determines the most appropriate approach for further characterization steps (Figure 11). After estimating the IC_50_ value by quantifying the inhibition percentage at different inhibitor concentrations, the assessed inhibitors can be classified as classic (i.e., IC_50_ >> [E]_0_) or tight binding (i.e., IC_50_ ≈ [E]_0_). Importantly, caution is recommended when comparing IC_50_ values, as they reflect not only the potency of the inhibitor but also the experimental conditions used for its estimation [139].

Step 2 consists of assessing the reversibility of the enzyme–inhibitor interaction. This property can be determined by the jump dilution assay [138,139,140] or alternative interaction-based techniques [35,36,141,142]. Although irreversible and pseudo-irreversible inhibitors exist in nature [143,144,145,146], most of the enzyme inhibitors described so far from marine organisms belong to the reversible class. Other relevant aspects of inhibition, such as association (i.e., time-dependent inhibition) and dissociation kinetics, are also investigated at this stage.

Elucidation of inhibition type (Step 3) and the estimation of Ki value (Step 4) are typically performed simultaneously by measuring enzyme velocity while varying substrate and inhibitor concentrations. The potency of competitive, noncompetitive, and uncompetitive inhibitors variate distinctively as substrate concentration increases [139], a fact exploited to identify their inhibition mode [138,140]. Once the inhibition modality has been elucidated, diverse mathematical approaches are available to estimate the Ki value for classical inhibitors [147,148,149]. On the other hand, determining Ki values for tight-binding inhibitors requires a specific methodology [150,151], as many of the assumptions applied to derive the equations for classic inhibitors are no longer valid in such systems.

The strategy is completed with a specificity validation step (Step 5) to discard unspecific actuators or PAINS (pan-assay interference compounds), which can act over many different proteins or interfere with bioassays through diverse mechanisms [152,153]. As these compounds can occur in nature [154], the specificity of novel inhibitors should be assessed against related and unrelated off-target reporters [155]. Additionally, experimental conditions can be conveniently manipulated (e.g., by adding to the reaction mix appropriate amounts of detergents such as Triton X-100) to minimize the occurrence of aggregators, the most common cause of artificial enzyme inhibition during screenings [155,156,157,158,159].

## 4. Concluding Remarks and Future Perspectives

During the last five decades, gorgonians and soft corals remained as an important source of new enzyme inhibitors, producing more than 50 reports and 127 new inhibitory molecules. Mainly, these inhibitors remain uncharacterized from an enzymological point of view; therefore, little speculation is possible regarding their potencies and inhibition mechanisms. As expected, low affinity inhibitors with IC_50_ values in the high micromolar range predominated (putatively) among them. Of note, the Acetylcholinesterase inhibitors Asperdiol (**29**) and 14-acetoxycrassine (**32**) [55], the PTP1B inhibitor (5′*Z*)-5-(2′,6′-Dimethylocta-5′,7′-dienyl)-furan-3-carboxylic acid (**18**) [49], the Phosphodiesterase-4 inhibitor 9,11,15-triacetoxy PGF2α methyl ester (**67**) [74], and the farnesyl protein transferase inhibitor 1*aR*,4*E*,8*E*,11*S*,11*aR*,14*aS*,14*bS*)-1*a*,5,9,14*a*-tetramethyl-12-methylene-13-oxo-1*a*,2,3,6,7,10, 11,11*a*,12,13,14*a*,14b-dodecahydrooxireno[2′,3′:13,14]cyclotetradeca[1,2-*b*]furan-11-yl acetate (**114**) [123], displaying preliminary potency estimates or Ki values in the sub- or low-micromolar range, stand outs as the most promising and clearly merit in-depth characterization studies. Interestingly, human target enzymes involved in complex pathways for chronic disorders such as PTP1B (type 2 diabetes and obesity) or the Ubiquitin–Proteasome system (neurodegenerative disorders, cancer, etc.) clearly captured the attention of researchers in the field during the period analyzed, with 38 inhibitors combined. Other enzymes, such as the HIV-1 protease or human Phosphodiesterase-4, also received special attention.

Due to the limited scope of this issue and for the sake of space, the presented strategy is not an exhaustive guide for the functional characterization of new inhibitors. In contrast, it aims to summarize relevant concepts and set a framework to address this goal, in practice burdensome enough to cause a halt in the study of potentially interesting enzyme inhibitors. The infrastructure required for performing most enzymatic reactions is readily available in many academic institutions. Numerous substrate and enzyme preparations are commercially available at affordable prices and basic enzymology is part of almost every formation program in biological sciences. In addition, many enzymes are considered important biomolecules for industrial, biomedical, and environmental applications, making their activity modulation by specific inhibitors a topic of particular interest. Paradoxically, in-depth enzymological characterization of inhibitors seems to be the prerogative of only a few laboratories, resulting in the accumulation of reports describing bioactivity prospection in natural extracts with no follow-up. This is particularly true for, but not exclusive to, organisms of marine origin, which have constituted an abundant source of enzyme inhibitors with unique properties.

Therefore, we hope this effort can contribute to democratizing and popularizing the procedures required in enzymological practice to minimally characterize, according to STRENDA suggestions, new enzyme inhibitors from marine organisms. In addition, this work might also help to propagate the message of the STRENDA project, focusing on the uniformity, quality, and completeness of experimental data, and hence reproducibility, in enzymological practice. Adherence to these standards should significantly facilitate relaunching of high-quality enzymological research in the context of marine organisms.

## Figures and Tables

**Figure 1 marinedrugs-21-00104-f001:**
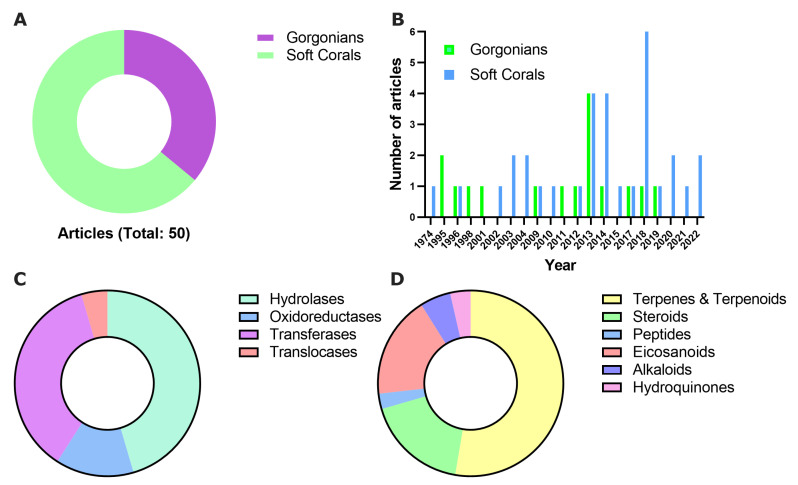
Distribution of the reviewed enzyme inhibitors isolated from soft corals and gorgonians. Inhibitors were classified according to their organism of origin (**A**), publication date (**B**), principal classes of targeted enzymes (**C**), and molecular structure (**D**).

**Figure 2 marinedrugs-21-00104-f002:**
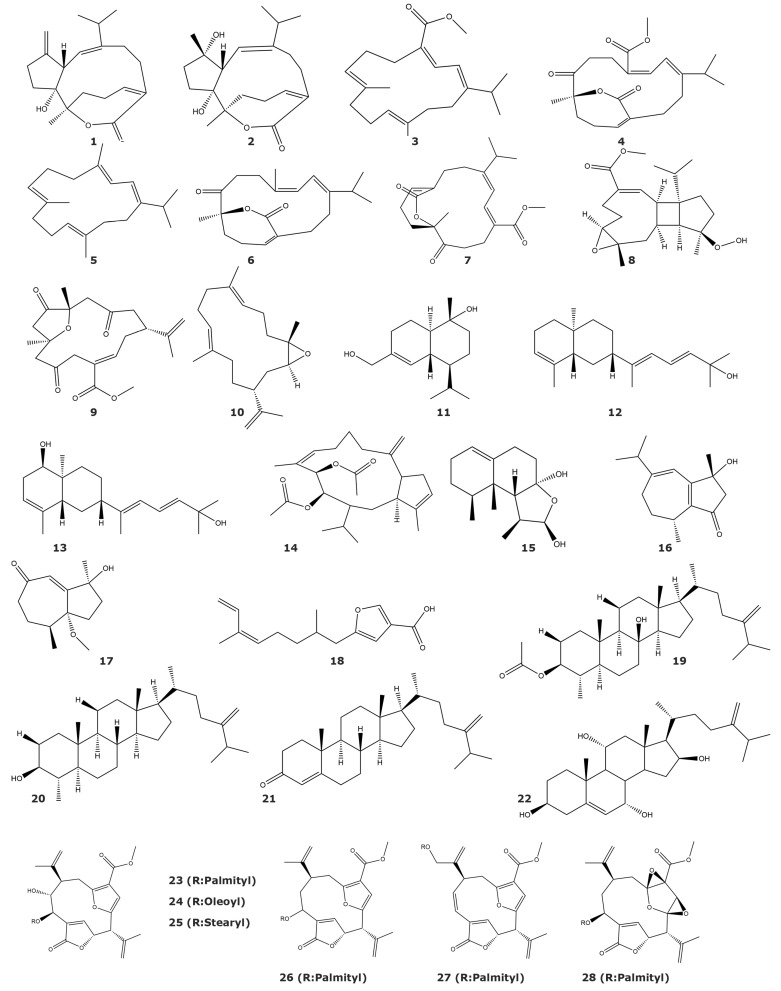
PTP1B inhibitors isolated from soft corals and gorgonians (compounds **1**–**28**).

**Figure 3 marinedrugs-21-00104-f003:**
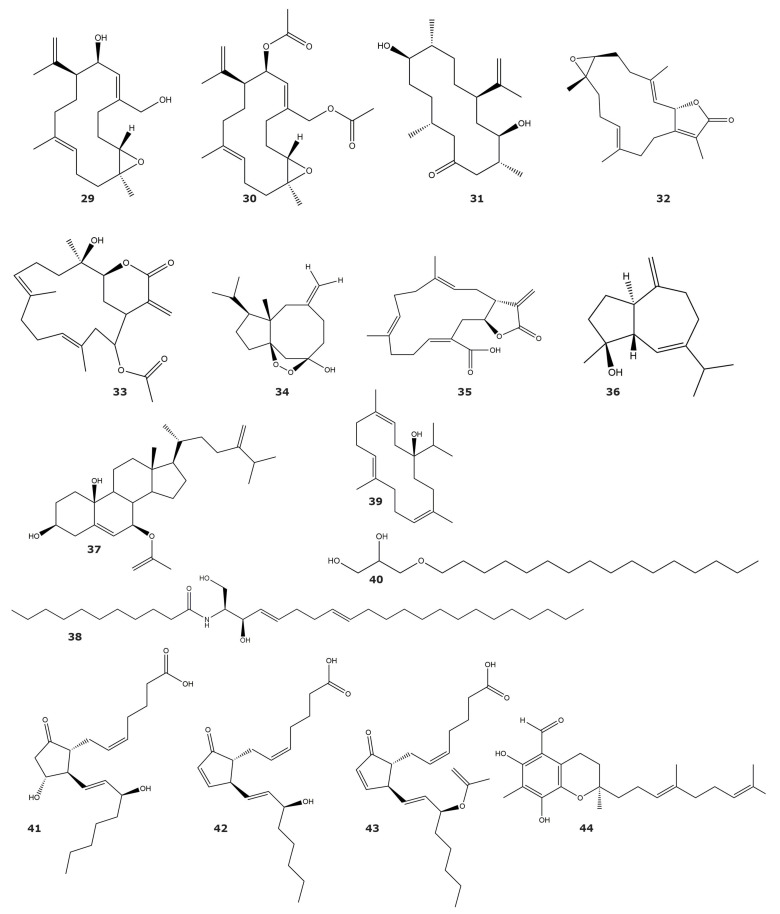
Hydrolase inhibitors isolated from soft corals and gorgonians (compounds **29**–**44**). Structures include inhibitors of acetylcholinesterase, HIV-1 protease, Elastase, and 3CLpro.

**Figure 4 marinedrugs-21-00104-f004:**
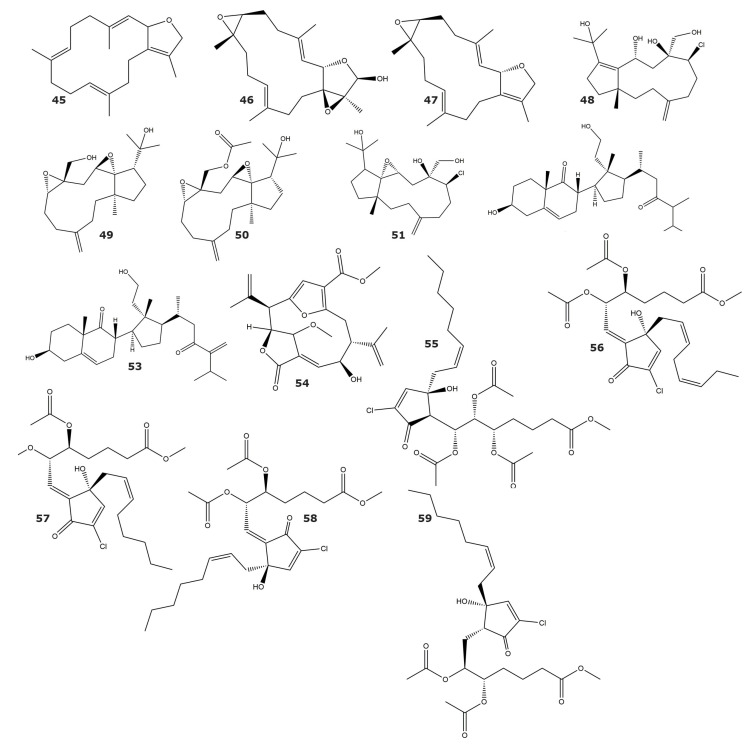
Inhibitors of the ubiquitin–proteasome system isolated from soft corals and gorgonians (compounds **45**–**59**).

**Figure 5 marinedrugs-21-00104-f005:**
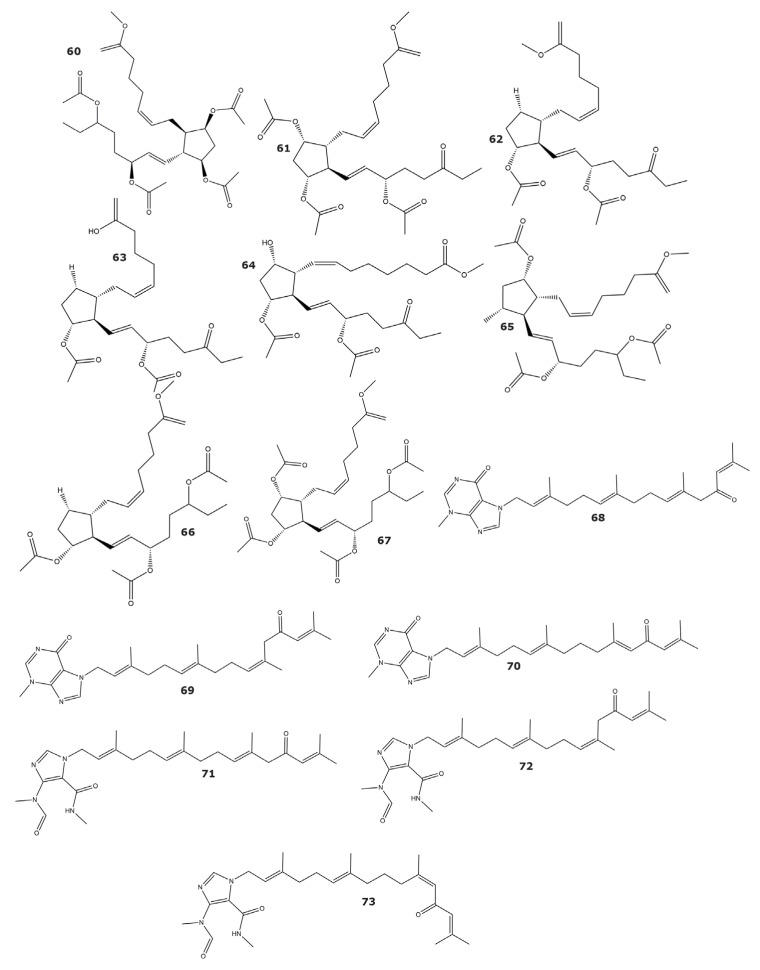
Phosphodiesterase-4 inhibitors isolated from soft corals and gorgonians (compounds **60**–**73**).

**Figure 6 marinedrugs-21-00104-f006:**
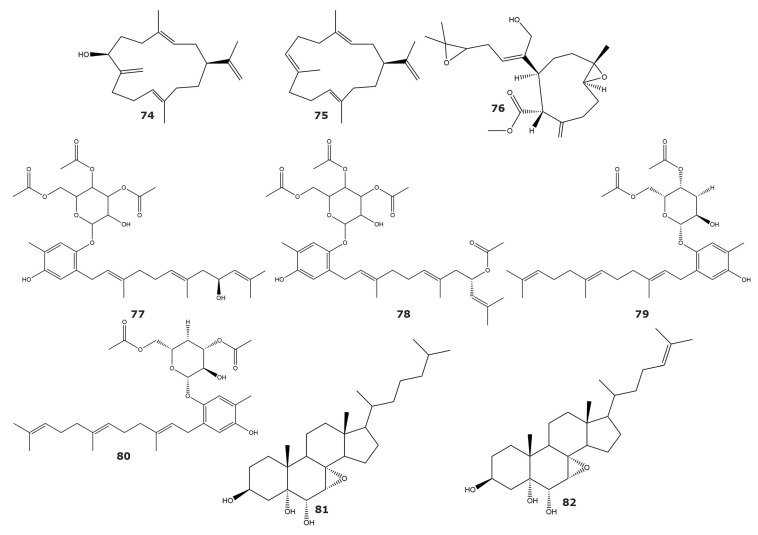
Hydrolase inhibitors isolated from soft corals and gorgonians (compounds **74**–**82**). Structures include inhibitors of α-glucosidase, Histone deacetylase 6 and Phospholipase A2.

**Figure 7 marinedrugs-21-00104-f007:**
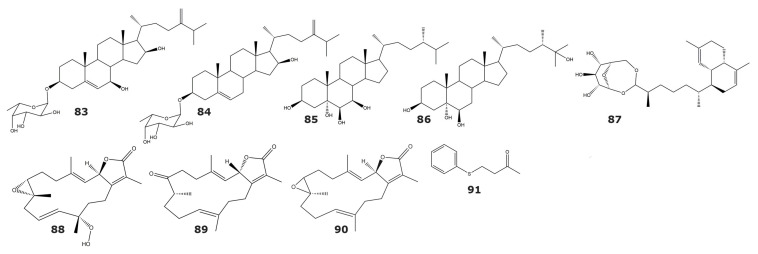
Oxidoreductase inhibitors isolated from soft corals and gorgonians (compounds **83**–**91**). Structures include inhibitors of 5α-Reductase, Cytochrome P450 1A, and Tyrosinase.

**Figure 8 marinedrugs-21-00104-f008:**
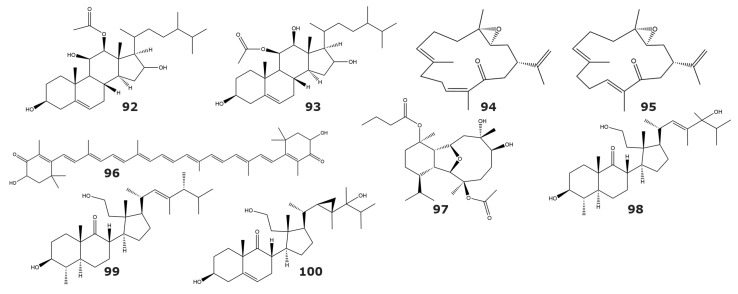
Transferase inhibitors isolated from soft corals and gorgonians (compounds **92**–**100**). Structures include inhibitors of Tyrosine kinase p56^lck^, IKKbeta kinase, EGFR kinase, and protein kinase C.

**Figure 9 marinedrugs-21-00104-f009:**
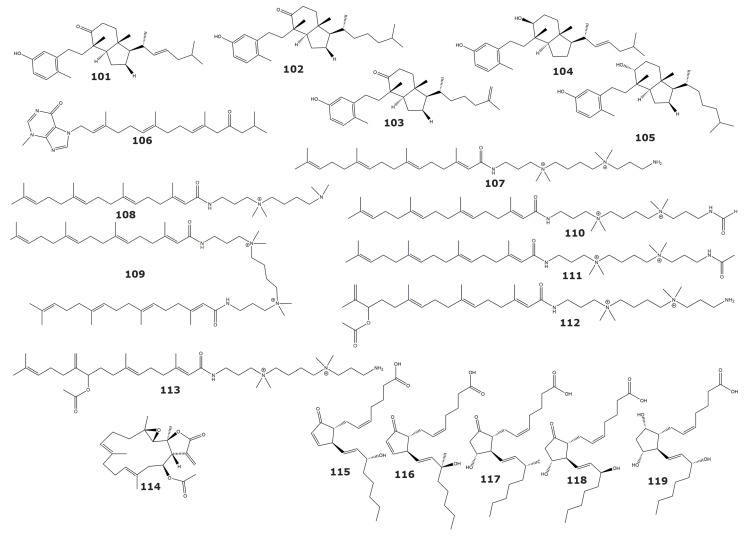
Transferase inhibitors isolated from soft corals and gorgonians (compounds (**101**–**119**). Structures include inhibitors of human tumor-related protein kinases, Casitas B-lineage lymphoma proto-oncogene B, Farnesyl Protein Transferase, and Glutathione S-transferase.

**Figure 10 marinedrugs-21-00104-f010:**
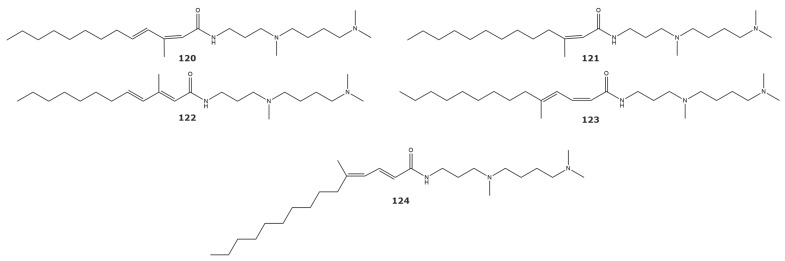
H(+)-pyrophosphatase inhibitors isolated from soft corals and gorgonians (compounds **120**–**124**).

**Figure 11 marinedrugs-21-00104-f011:**
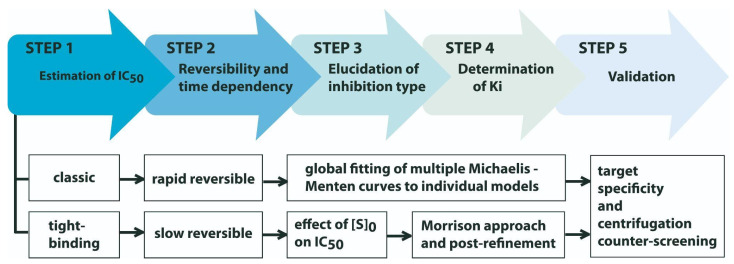
Proposed strategy for the characterization of reversible enzyme inhibitors from marine organisms according to STRENDA guidelines. The strategy is presented as a flowchart with five sequential steps. As alternative approaches or methodologies are possible in each step, two specific sequences are proposed according to the properties of the inhibitor (e.g., classic, rapid reversible inhibitor vs. tight-binding, slow reversible inhibitor).

**Table 1 marinedrugs-21-00104-t001:** List of enzymes targeted by the inhibitors isolated from soft corals and gorgonians. Associated diseases and/or processes are also indicated.

Enzyme	EC Number	Class	Disease/Process
Protein tyrosine phosphatase 1B	3.1.3.48	Hydrolase	Cancer
Acetylcholinesterase	3.1.1.7	Hydrolase	Alzheimer’s disease
HIV-1 protease	3.4.23.16	Hydrolase	HIV-AIDS
Elastase	3.4.21.11	Hydrolase	Immune response/inflammation
3CLpro Enzyme	3.4.22.69	Hydrolase	COVID-19
Ubiquitin-proteasome system	3.4.25.1	Hydrolase	Neurodegenerative/immune-related/cancer
Phosphodiesterase-4	3.1.4.53	Hydrolase	Inflammatory/asthma/chronic obstructive pulmonary diseases
α-glucosidase	3.2.1.207	Hydrolase	Type II diabetes
Histone deacetylase 6	3.5.1.98	Hydrolase	Immunoregulation
Phospholipase A2	3.1.1.4	Hydrolase	Cancer/inflammation/atherosclerosis
5α-Reductase	1.3.1.22	Oxidoreductase	Benign hyperplasia/male pattern baldness
Cytochrome P450 1A	1.14.14.1	Oxidoreductase	Cancer
Tyrosinase	1.14.18.1	Oxidoreductase	Development of skin-whitening agents
Tyrosine kinase p56^lck^	2.7.10.2	Transferase	Inflammatory/autoimmune disorders
IKKbeta kinase	2.7.11.10	Transferase	Inflammatory diseases
Epidermal growth factor receptor kinase	2.7.10.1	Transferase	Cancer
Protein kinase C	2.7.11.13	Transferase	Cancer/neurological/cardiovascular disorders
Human tumor-related protein kinases	-	Transferase	Cancer
Casitas B-lineage lymphoma proto-oncogene B	2.3.2.27	Transferase	Cancer
Farnesyl protein transferase	2.5.1.58	Transferase	Cancer
Glutathione S-transferase	2.5.1.18	Transferase	Cancer
H (+)-pyrophosphatase	7.1.3.1	Translocase	-

**Table 2 marinedrugs-21-00104-t002:** PTP1B inhibitors isolated from soft corals and gorgonians.

Compound	IC_50_ (μM)	Source	Classification	Reference
Sarsolides A (**1**)	6.8	*Sarcophyton trocheliophorum Marenzeller*	Soft coral	[43]
Sarsolides B (**2**)	27.1			
Sarcophytonolide N (**3**)	5.95			[44]
Sarcrassin E (**4**)	6.33			
Cembrene C (**5**)	26.6			
4*Z*,12*Z*,14*E*-sarcophytolide (**6**)	15.4			
Ketoemblide (**7**)	27.2			
Methyl sarcotroates B (**8**)	6.97			[45]
Sinulin D (**9**)	47,500	*Sinularia* sp.	Soft coral	[46]
(1*R*,3*S*,4*S*,7*E*,11*E*)-3,4-epoxycembra-7,11,15-triene (**10**)	12,500			
15-hydroxy-α-cadinol (**11**)	22,100			
Sinupol (**12**)	63.9	*Sinularia polydactyla*	Soft coral	[47]
(1*R*,4*aR*,6*S*,8*aS*)-6-((2*E*,4*E*)-6-hydroxy-6-methylhepta-2,4-dien-2-yl)-4,8*a*-dimethyl-1,2,4*a*,5,6,7,8,8*a*-octahyronaphthalen-1-ol (**13**)	75.5			
Sinulacetate (**14**)	51.8			
Linardosinene I (**15**)	10.67 ^1^	*Litophyton nigrum*	Soft coral	[48]
Molestins C (**16**)	218	*Sinularia* cf. *molesta*	Soft coral	[49]
Molestins D (**17**)	344			
(5′*Z*)-5-(2′,6′-Dimethylocta-5′,7′-dienyl)-furan-3-carboxylic acid (**18**)	1.24			
(3β,4α,5α,8β)-4-methylergost-24(28)-ene-8-ol-3-monoacetate (**19**)	22.7	*Sinularia depressa*	Soft coral	[50]
(3β, 4α, 5α)-4-methylergost-24(28)-ene-3-ol (**20**)	19.5			
Ergost-4,24(28)-diene-3-one (**21**)	15.3			
7α-hydroxy-crassarosterol A (**22**)	33.05	*Sinularia flexibilis*	Soft coral	[51]
Lipidyl pseudopterane A (**23**)	71 ^1^	*Pseudopterogorgia acerosa*	Gorgonian	[52]
Lipidyl pseudopterane B (**24**)	ND			
Lipidyl pseudopterane C (**25**)	ND			
Lipidyl pseudopterane D (**26**)	ND			
Lipidyl pseudopterane E (**27**)	ND			
Lipidyl pseudopterane F (**28**)	ND			

^1^ μg/mL, ND—not determined.

**Table 3 marinedrugs-21-00104-t003:** Inhibitors of the hydrolases acetylcholinesterase, HIV-1 protease, elastase, and 3CLpro isolated from soft corals and gorgonians.

Compound	IC_50_ (μM)	Source	Classification	Reference
*Acetylcholinesterase*				
Asperdiol (**29**)	0.358	*Eunicea knighti*	Soft coral	[55]
Asperdiol diacetate (**30**)	ND			
8*R*-dihydroplexaurolone (**31**)	ND			
14-acetoxycrassine (**32**)	1.40	*Pseudoplexaura porosa*	Gorgonian	
Sarcophine (**33**)	ND	*Sarcophyton glaucum*	Soft coral	[56]
Sinuketal (**34**)	ND	*Sinularia* sp.	Soft coral	[46]
Crassumolide E (**35**)	ND	*Lobophytum* sp.	Soft coral	[57]
*HIV-1 protease*				
Alismol (**36**)	7.20	*Litophyton arboreum*	Soft coral	[58]
7β-acetoxy-24-methylcholesta-5-24(28)-diene-3,19-diol (**37**)	4.85			
erythro-N-dodecanoyl-docosasphinga-(4*E*,8*E*)-dienine (**38**)	4.80			
Sarcophytol M (**39**)	15.7			
Chimyl alcohol (**40**)	26.6			
*Elastase*				
PcKuz1	ND	*Palythoa caribaeorum*	Soft coral	[59]
PcKuz2	ND			
PcKuz3	ND			
(15R)-PGE2 (**41**)	ND	*Plexaura homomalla*	Gorgonian	[60]
(15R)-PGA2 (**42**)	ND			
(15*R*)-*O*Ac-PGA2 (**43**)	ND			
*3CLpro*				
Tuaimenal A (**44**)	21	*Duva florida*	Soft coral	[61]

ND—not determined.

**Table 4 marinedrugs-21-00104-t004:** Inhibitors of the hydrolases ubiquitin–proteasome system and phosphodiesterase-4, isolated from soft corals and gorgonians.

Compound	IC_50_ (μM)	Source	Classification	Reference
*Ubiquitin–Proteasome system*				
Sarcophytonin A (**45**)	ND	*Sarcophyton trocheliophorum*	Soft coral	[70]
Laevigatol A (**46**)	ND			
Sarcophytoxide (**47**)	ND	*Sarcophyton ehrenbergi*	Soft coral	
Sarcophine (**33**)	ND			
Clavinflol C (**48**)	ND	*Clavularia flava*	Soft coral	[71]
Stolonidiol (**49**)	ND			
Stolonidiol-17-acetate (**50**)	ND			
Clavinflol B (**51**)	ND			
3β,11-dihydroxy-24-methyl-9,11-secocholest-5-en-9,23-dione (**52**)	ND			
3β,11-dihydroxy-24-methylene-9,11-secocholest-5-en-9,23-dione (**53**)	ND			
Methyl (2*R*,3*S*,8*S*,9*R*,*E*)-8-hydroxy-15-methoxy-5-oxo-2,9-di(prop-1-en-2-yl)-4,14-dioxatricyclo [9.2.1.1^3,6^] pentadeca-1(13),6,11-triene-12-carboxylate (**54**)	9.77	*Pseudopterogorgia acerosa*	Gorgonian	[72]
PNG 2 (**55**)	ND	*Telesto riisei*	Soft coral	[73]
PNG 3 (**56**)	ND			
PNG 4 (**57**)	ND			
Z-PNG 4 (**58**)	ND			
PNG 6 (**59**)	ND			
*Phosphodiesterase-4*				
Sarcoehrendin B (**60**)	3.7	*Sarcophyton ehrenbergi*	Soft coral	[74]
Sarcoehrendin D (**61**)	10.6			
Sarcoehrendin F (**62**)	12.1			
Sarcoehrendin H (**63**)	16.9			
Sarcoehrendin J (**64**)	7.2			
9α,15α-diacetoxy-11α-hydroxy-5*Z*,13*E*-prostadienoic acid methyl ester (**65**)	4.7			
(5*Z*,9α,11α,13*E*,15*S*)-11,15-bis(acetoxy)-9-hydroxyprosta-5,13-dien-1-oic acid methyl ester (**66**)	5.5			
9,11,15-triacetoxy PGF2α methyl ester (67)	1.4			
Malonganenone L (**68**)	8.5	*Echinogorgia pseudossapo*	Gorgonian	[75]
Malonganenone M (**69**)	ND			
Malonganenone N (**70**)	ND			
Malonganenone O (**71**)	ND			
Malonganenone P (**72**)	ND			
Malonganenone Q (**73**)	20.3			

ND—not determined.

**Table 5 marinedrugs-21-00104-t005:** Inhibitors of hydrolases α-glucosidase, histone deacetylase 6, and phospholipase A2, isolated from soft corals and gorgonians.

Compound	IC_50_ (μM)	Source	Classification	Reference
*α-glucosidase*				
Sinulacrassin B (**74**)	10.65	*Sinularia crassa*	Soft coral	[81]
S-(+)-cembrane A (**75**)	30.31			
*Histone deacetylase 6*				
Methyl (1*S*, 4*S*, 5*R*, 9*S*9-4-((*Z*)-4-(3,3-dimethyloxiran-2-yl-)-1-hydrioxybut-2-en-2-yl)-1-methyl-6-methylene-10-oxabicyclo[7.1.0]decane-5-carboxylate (**76**)	80	*Xenia elongata*	Soft coral	[82]
*Phospholipase A2*				
Euplexide A (**77**)	ND	*Euplexaura anastomosans*	Gorgonian	[83]
Euplexide B (**78**)	ND			
Euplexide F (**79**)	ND			[84]
Euplexide G (**80**)	ND			
7*α*,8*α*-epoxy-3*β*,5*α*,6*α*-trihydroxycholestane (**81**)	ND	*Acabaria undulata*	Gorgonian	[85]
24-methyl-7*α*,8*α*-epoxy-3*β*,5*α*,6*α*-trihydroxycholest-22-ene (**82**)	ND			

ND—not determined.

**Table 6 marinedrugs-21-00104-t006:** Inhibitors of oxidoreductases isolated from soft corals and gorgonians.

Compound	IC_50_ (μM)	Source	Classification	Reference
*5α-Reductase*				
24-methylenecholest-5-ene-3β,7β, 16β-triol-3-O-α-l-fucopyranoside (**83**)	ND	*Sinularia crassa Tixier–Durivaul*	Soft coral	[95]
		*Sinularia gravis Tixier–Durivault*	Soft coral	
		*Sinularia* sp.	Soft coral	
24-methylenecholest-5-ene-3β,7β, 16β-diol-3-O-α-l-fucopyranoside (**84**)	ND	*Sinularia* sp.	Soft coral	
		*Cladiella* sp.	Soft coral	
(24*S*)-24-methylcholestane-3β,5α,6β,7β-tetrol (**85**)	ND	*Lobophytum crassum*	Soft coral	
(24*S*)-24-methylcholestane-3β,5α,6β,25-tetrol (**86**)	ND	*Lobophytum* sp.	Soft coral	
Lemnabourside (**87**)	250	*Nephthea chabroli*	Soft coral	[96]
*Cytochrome P450 1A*				
12(S)-hydroperoxylsarcoph-10-ene, 8-epi-sarcophinone (**88**)	*2.7*	*Sarcophyton glaucum*	Soft coral	[97]
8-epi-sarcophinone (**89**)	3.7			
Ent-sarcophine (**90**)	3.4			
	*Tyrosinase*			
4-(phenylsulfanyl)butan-2-one (**91**)	34.5 *	*Cladiella australis*	Soft coral	[98]

* K_i_._,_ ND—not determined.

**Table 7 marinedrugs-21-00104-t007:** Inhibitors of the transferases tyrosine kinase p56^lck^, IKKbeta kinase, EGFR kinase, and protein kinase C, isolated from soft corals and gorgonians.

Compound	IC_50_ (μM)	Source	Classification	Reference
*Tyrosine kinase p56^lck^*				
12β-acetoxyergost-5-ene-3β,11β,16-triol (**92**)	ND	*Capnella lacertiliensis*	Soft coral	[104]
11β-acetoxyergost-5-ene-3β,12β,16-triol (**93**)	ND			
*IKKbeta kinase*				
3,4-epoxy,13-oxo,7*E*,11*Z*,15-cembratriene (**94**)	ND	*Sarcophyton* sp.	Soft coral	[105]
3,4-epoxy,13-oxo,7*E*,11*E*,15-cembratriene (**95**)	ND			
Astaxanthin (**96**)	ND	*Subergorgia* sp.	Gorgonian	
*EGFR kinase*				
Pachycladin A (**97**)	ND	*Cladiella pachyclados*	Soft coral	[106]
*Protein kinase C (PKC)*				
2*S*,4*aS*,5*S*,6*S*,8*aS*)-6-hydroxy-2-((1*S*,2*R*,3*R*)-3-((2*R*,*E*)-5-hydroxy-4,5,6-trimethylhept-3-en-2-yl)-2-(2-hydroxyethyl)-2-methylcyclopentyl)-5,8*a*-dimethyloctahydronaphthalen-1(2H)-one (**98**)	NA	*Pseudopterogorgia* sp.	Gorgonian	[107]
(2*S*,4*aS*,5*S*,6*S*,8*aS*)-6-hydroxy-2-((1*S*,2*R*,3*R*)-2-(2-hydroxyethyl)-2-methyl-3-((2*R*,5*R*,*E*)-4,5,6-trimethylhept-3-en-2-yl)cyclopentyl)-5,8a-dimethyloctahydronaphthalen-1(2H)-one (**99**)	NA			
3β, 11, 24-trihidroxy, 9,11-secogorgos-5-en-9-one (**100**)	NA			

ND—not determined, NA—not available.

**Table 8 marinedrugs-21-00104-t008:** Inhibitors of the transferases human tumor-related protein kinases, casitas B-lineage lymphoma proto-oncogene B (Cbl-b), farnesyl protein transferase, and glutathione S-transferase, isolated from soft corals and gorgonians.

Compound	IC_50_ (μM)	Source	Classification	Reference
*Human tumor-related protein kinases*				
Calicoferol A (**101**)	9.33 (ALK), 38.1 (Aurora-B), 21.9 (AXL), 16.9 (FAK), 3.16 (IGF-1R), 34.0 (MET wt), 2.40 (SRC), 4.95 (VEGF-R2)	*Astrogorgia* sp.	Gorgonian	[119]
Calicoferol E (**102**)	4.14 (ALK), 14.7 (AXL), 9.92 (FAK), 2.42 (IGF-1R), 47.7 (MET wt), 2.24 (SRC), 4.60 (VEGF-R2)			
24-exomethylenecalicoferol E (**103**)	4.36 (ALK), 20.2 (AXL), 10.7 (FAK), 2.30 (IGF-1R), 27.5 (MET wt), 1.48 (SRC), 4.85 (VEGF-R2)			
9β-hydroxy-9,10-secosteroid astrogorgol F (**104**)	4.73 (ALK), 32.6 (AXL), 9.63 (FAK), 2.46 (IGF-1R), 71.5 (MET wt), 2.17 (SRC), 6.01 (VEGF-R2)			
9α-hydroxy-9,10-secosteroid astrogorgiadiol (**105**)	7.55 (ALK), 25.1 (Aurora-B), 16.9 (AXL), 13.2 (FAK), 2.77 (IGF-1R), 48.9 (MEK1 wt), 78.0 (MET wt), 1.91 (SRC), 4.35 (VEGF-R2)			
Malonganenone D (**106**)	ND	*Euplexaura robusta*	Gorgonian	[120]
*Casitas B-lineage lymphoma proto-oncogene B (Cbl-b)*				
Sinularamide A (**107**)	ND	*Sinularia* sp.	Soft coral	[122]
Sinularamide B (**108**)	ND			
Sinularamide C (**109**)	6.5			
Sinularamide D (**110**)	ND			
Sinularamide F (**111**)	ND			
Sinularamide G (**112**)	ND			
*Farnesyl Protein Transferase*				
1*aR*,4*E*,8*E*,11*S*,11*aR*,14*aS*,14*bS*)-1a,5,9,14*a*-tetramethyl-12-methylene-13-oxo-1*a*,2,3,6,7,10,11,11*a*,12,13,14*a*,14b-dodecahydrooxireno[2′,3′:13,14]cyclotetradeca[1,2-*b*]furan-11-yl acetate (**113**)	0.15	*Lobophytum cristagalli*	Soft coral	[123]
*Glutathione S-transferase*				
15(*S*)-PGA2 (**114**)	75.4	*Plexaura homomalla* ^1^	Gorgonian	[124]
15(*R*)-15-methyl PGA2 (**115**)	136.7			
15(*S*)-PGE2 (**116**)	312.2			
15(*R*)-PGE2 (**117**)	ND			
15(*S*)-PGF2*a* (**118**)	334.6			

^1^ Commercial compounds that represent the major classes of prostaglandins present in *P. homomalla*. ND—not determined.

## Data Availability

Not applicable.

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
