# Peer review of "Enzyme Inhibitors from Gorgonians and Soft Corals"

_marinedrugs, 2023, doi:10.3390/md21020104_

Round 1

Reviewer 1 Report

The manuscript entitled “Enzyme inhibitors from gorgonian and soft coral” is interesting. In this manuscript, the authors describe enzyme inhibitors isolated from gorgonian and soft coral by searching scientific literature from 1974 to 2022. This manuscript could be accepted after a minor revision according to the comments below.

 Minor comments:

(1)   Page 2, line 48 “----------------- and environmental applications [Ref?]” reference is needed.

(2)   Table 1 and rest---. It would be easy to understand for the readers if the authors could show soft corals and gorgonian separately.

(3)   The paper will be more informative and helpful for the readers if the authors can show the application of enzymes in a separate table which have already been isolated from 1974 to 2022.

The paper is too long (42 pages). Usually, readers like brief and key but informative descriptions. I would suggest to the authors to switch some tables and figures to suppl. info. 

Author Response

Please, find below our answers (colored in red) to the reviewer’s comments.

Comments:
The manuscript entitled “Enzyme inhibitors from gorgonian and soft coral” is interesting. In this  manuscript, the authors describe enzyme inhibitors isolated from gorgonian and soft coral by searching scientific literature from 1974 to 2022. This manuscript could be accepted after a minor revision according to the comments below.
Thank you for your valuable and positive comments on our manuscript.
Minor comments:
(1) Page 2, line 48 “----------------- and environmental applications [Ref?]” reference is needed.
We have added updated references on the matter. See refs. 14 and 15 in the revised version.
(2) Table 1 and rest---. It would be easy to understand for the readers if the authors could show soft  corals and gorgonian separately.
Thanks for this comment. Both reviewers suggested modifications to the content and distribution of tables (Please, see comment 2 of Reviewer 2). Although we agree with the spirit of that, we didn't find a system to completely satisfies the suggestions of both reviewers (tables by classification and by enzymes); as it would result in more than 30 tables (some of them with very few entries). Therefore, we decided to assemble 8 tables, and to include the column "Classification" to indicate whether the source was a soft coral or a gorgonian. Finally, we also generated a supplementary excel file containing the info for the 127 inhibitors on the manuscript, with separate sheets for soft corals and 
gorgonians. For interested readers, this resource would facilitate data exploration better than tables, as it supports combined searches (e.g., "Hydrolases" AND "terpenes&terpenoids"). 
(3) The paper will be more informative and helpful for the readers if the authors can show the application of enzymes in a separate table which have already been isolated from 1974 to 2022.
As requested by the reviewer, we have now included a new table summarizing the applications of the enzymes targeted by the described inhibitors. Please, see Table 1, pages 3-4 in the revised version of the manuscript. 
(4) The paper is too long (42 pages). Usually, readers like brief and key but informative descriptions. 
I would suggest to the authors to switch some tables and figures to suppl. info.
Both reviewers agreed on this appreciation. To make the manuscript more readable and briefer, we have condensed sections 3.1 - 3.6 (7 pages) of the previous version to one section (less than 2 pages) in the revised manuscript. 

Reviewer 2 Report

The authors did a comprehensive survey and summarized enzyme inhibitors from gorgonians and soft corals. In addition, they included standard guidelines for evaluating the inhibition activities. However, appropriate revisions are needed to improve the manuscript before publication.

 1.     Sections 3.1 to 3.6 might not be suitable to be included in this manuscript. Alternatively, the authors could make a concise table summarizing the information and references. Or it could be attached as “Supporting Information”. The manuscript should have a focus as its title.

2.     I suggest the authors place the activity data and chemical structures in the corresponding parts of the manuscript. Related tables and figures should be split and placed in the corresponding paragraph.

3.     “which contained more than 120 inhibitor molecules”, please add the exact number.

  1. Page 1, lines 19–22, the “enzyme inhibitors” is not a structure type as terpenoids and steroids. Please rewrite this sentence.
  2. In the manuscript, the compound numbers should be attached behind the compound name, which will be more convenient for readers to find the structures of compounds.
  3. In figure 7, the compound name should be replaced by the compound number.
  4. Page 2, line 48, please attach some references to support the environmental applications of natural enzyme inhibitors.
  5. For compound names, the “R, S, E, Z, a, b” should use the italic font.
  6. Page 4, line 142, “Sinularia cf. molesta” should be “Sinularia cf. molesta”, and all the specie names of gorgonians and soft corals should use the italic font.

10.  More careful proofreading is needed. Too many obvious grammar mistakes or typos could be found throughout the manuscript—lines 23, 147, 401, 440, etc.

Author Response

Please, find below our answers (colored in red) to the reviewer’s comments.

Reviewer: 2
Comments:
The authors did a comprehensive survey and summarized enzyme inhibitors from gorgonians and soft corals. In addition, they included standard guidelines for evaluating the inhibition activities. However, appropriate revisions are needed to improve the manuscript before publication.
We thank the reviewer for the thorough revision of our manuscript and appreciate the valuable suggestions to improve it further.

1. Sections 3.1 to 3.6 might not be suitable to be included in this manuscript. Alternatively, the authors could make a concise table summarizing the information and references. Or it could be attached as “Supporting Information”. The manuscript should have a focus as its title.
A similar comment was also raised by Reviewer 1. As mentioned before, we have substantially reduced the indicated sections in the revised manuscript. The condensed version still provides valuable information and key references on the steps required for characterizing new enzyme inhibitors.
2. I suggest the authors place the activity data and chemical structures in the corresponding parts of the manuscript. Related tables and figures should be split and placed in the corresponding paragraph.
As mentioned before (please, see comment 2 of Reviewer 1), the number of enzymes (37) included in the manuscript is too high for splitting tables and figures accordingly. However, we have fragmented tables (from 4 to 8 in the revised version) and figures (from 8 to 11 in the new version) to place them as close as possible to their reference paragraph, as requested by the reviewer. We have also generated a supplementary excel file to facilitate data exploration. 
3. “which contained more than 120 inhibitor molecules”, please add the exact number.
We have now included in the revised manuscript the exact number (127) of inhibitor molecules, as requested by the reviewer.
4. Page 1, lines 19–22, the “enzyme inhibitors” is not a structure type as terpenoids and steroids. Please rewrite this sentence.
Thanks for this observation. We agree on that enzyme inhibitors are structurally too heterogeneous to be considered a structure type, such as terpenoids or steroids. In the indicated sentence, however, the term "enzyme inhibitors" refers to a functional class, not to a structural one. To avoid possible confusion in this regard, we have added the word “functional” (i.e., "… However, enzyme inhibitors, a functional class of bioactive compounds…") to clarify this issue in the revised manuscript. 
5. In the manuscript, the compound numbers should be attached behind the compound name, which will be more convenient for readers to find the structures of compounds.
As requested by the reviewer, we have now included the number of the compounds behind their names through the revised manuscript.
6. In figure 7, the compound name should be replaced by the compound number.
Thanks for this comment. For uniformity, we have modified the indicated Figure to include the number of compounds instead of their names in the new version. Please, see Figure 10, page 25 in the revised manuscript.
7. Page 2, line 48, please attach some references to support the environmental applications of natural enzyme inhibitors.
As mentioned earlier, we have now added updated references on the matter. See refs. 14 and 15 in the revised manuscript.
8. For compound names, the “R, S, E, Z, a, b” should use the italic font.
Thanks for this comment. We have corrected these format errors in the revised version of the manuscript.
9. Page 4, line 142, “Sinularia cf. molesta” should be “Sinularia cf. molesta”, and all the specie names of gorgonians and soft corals should use the italic font.
As requested by the reviewer, we have now corrected the format of the indicated word and used italic 
fonts for the names of gorgonians and soft corals in the revised manuscript.
10. More careful proofreading is needed. Too many obvious grammar mistakes or typos could be found throughout the manuscript—lines 23, 147, 401, 440, etc.
We are sorry for that. We have now corrected the mentioned mistakes/typos and have performed an extensive revision of the text to improve the English language and style in the revised manuscript.

Round 2

Reviewer 1 Report

The authors nicely addressed all of my comments and revised accordingly. However, I still have concerns regarding the length of the manuscript. As I mentioned in my previous review, to attract readers the authors could switch some data to supplementary info. Readers who are interested in knowing more details could visit the supplementary info.